

# Regional and seasonal evolution of melt ponds on Arctic sea ice

Hannah Niehaus[1], Gunnar Spreen[1], Larysa Istomina[2], and Marcel Nicolaus[2]

[1]Institute of Environmental Physics, University of Bremen, Bremen, Germany
[2]Alfred Wegener Institute, Helmholtz Centre for Polar and Marine Research, Bremerhaven, Germany

**Correspondence:** Gunnar Spreen (gunnar.spreen@uni-bremen.de)

**Abstract.** Melt ponds on sea ice significantly modify the absorption of solar radiation by the sea ice-ocean system in the Arctic, thereby influencing the energy budget and sea ice mass balance. Consequently, melt ponds are crucial to the positive sea ice-albedo feedback mechanism, which is a major factor to the enhanced Arctic warming observed in recent decades, with implications for the global climate. Given the high seasonal and interannual variability of melt ponds, understanding the mechanisms behind their evolution and their impact on the sea ice state is essential for improving sea ice and global climate models. This analysis must also take into account regional differences in melt pond evolution.

Here we present seven years (2017–2023) of melt pond fraction data produced with the Melt Pond Detection 2 (MPD2) algorithm from optical Sentinel-3 satellite observations. We demonstrate strong regional differences in the melt pond evolution process as well as high seasonal and interannual variability. The study shows that the variability is lower in the Central Arctic than in the marginal Arctic seas, which is in compliance with the more stable sea ice coverage in the Central Arctic. Hence this region also shows the highest potential of using melt pond fractions at the beginning of summer as an indicator for the summer surface energy budgets and thus the progress of melt season. Between the nine regions for the marginal seas, strong differences in melt pond variability are observed.

Sea ice surface topography and air temperature are investigated as primary factors to influence melt pond formation and evolution. Air temperature shows an immediate impact on the melt pond coverage, whose short-lived changes can be well resolved with the new MPD2 melt pond fraction product. A higher sea ice surface roughness leads to lower melt pond fractions in the beginning of the season. Later in the melt season, different behavior of melt pond drainage leads to a reversal of that relationship and hence lower melt pond fractions are observed on the level, flatter sea ice.

## 1 Introduction

The summer period in the central Arctic is characterized by the continuous presence of shortwave solar radiation. In addition to the infrared terrestrial radiation, which mainly is determined by the presence of clouds, this shortwave radiation can rapidly warm the air and the surface of the sea ice. The sea ice cover acts as a protective shield against incoming solar radiation due to its high albedo, limiting the heat uptake of the Arctic Ocean. Thick, white ice or even snow-covered sea ice reflects more than 70 % of the incoming radiation (Malinka et al., 2018; Light et al., 2022), whereas the darker, open ocean would absorb 90 % of the solar energy if not protected by sea ice (Pohl et al., 2020). Consequently, changes in sea ice extent, thickness or



surface albedo have a significant impact on the Arctic energy budget and sea ice mass balance (Fetterer and Untersteiner, 1998; Perovich et al., 2002a; Nicolaus et al., 2012).

Melt ponds, which form on the surface of Arctic sea ice from surface melt water during summer, drastically reduce the surface albedo (Eicken et al., 2004; Istomina et al., 2015a; Light et al., 2022). The absorption of solar energy in the sea ice and
upper ocean (Perovich et al., 2003) is enhanced, which leads to a warming and accelerates the sea ice melt. This self-enhancing feedback loop is known as the sea ice-albedo feedback mechanism (Curry et al., 1995; Perovich et al., 2008; Wendisch et al., 2023). Due to this feedback mechanism, the presence of melt ponds have a strong impact on the sea ice properties, stability and disintegration (Kwok and Untersteiner, 2011; Holland et al., 2012). Some studies even suggest that melt ponds could be used to predict the seasonal Arctic minimum sea ice extent in September (Schröder et al., 2014; Liu et al., 2015), which has
decreased dramatically in recent decades (Stroeve et al., 2012b, a; Screen, 2021).

Melt ponds also increase the transmissivity of the sea ice below them and thereby the light availability under the sea ice. This impacts the ecology under the ice and is one of the examples of direct coupling between the physical and biological sea ice systems (Tsamados et al., 2015; Katlein et al., 2019; Stroeve et al., 2024). The increased transmissivity for melt ponds also enhances the upper ocean warming and contributes to the sea ice albedo feedback (Perovich et al., 2007; Wendisch et al.,
2023). Given the importance of melt ponds for the sea ice evolution and hence the Arctic climate, there have been efforts to produce pan-Arctic data sets of melt pond coverage from satellite observations (Rösel et al., 2012; Zege et al., 2015; Ding et al., 2020; Lee et al., 2020; Feng et al., 2022; Peng et al., 2022) to enable the analysis of their large-scale distribution. This is much needed as melt pond evolution processes have been shown to be of high spatial and temporal variability (Scharien and Yackel, 2005; Polashenski et al., 2012; Tao et al., 2024) that is not yet adequately considered in global climate models (Flocco
and Feltham, 2007; Hunke et al., 2013; Dorn et al., 2018; Diamond et al., 2024). In addition, the quantification of the influence of melt ponds on the surface radiation budget would benefit from a regional and pan-Arctic analysis of melt pond evolution, which is fundamental for a deeper understanding of the amplified warming observed in the Arctic in recent decades (Serreze et al., 2009; Wendisch et al., 2023).

In contrast to other melt pond products based on machine learning approaches, Zege et al. (2015) have developed an entirely
physical algorithm, called MPD1, that retrieves pan-Arctic melt pond fractions on sea ice from optical satellite observations. Niehaus et al. (2024) have presented the new MPD2 version of this algorithm for Sentinel-3 OLCI data that includes the open ocean as a third surface type class along with the melt pond and sea ice classes, by using prior information to support and constrain the physical model. This approach has significantly improved the reliability of melt pond products on Arctic sea ice.

Here we apply the MPD2 algorithm on seven years of satellite data from 2017 to 2023 to produce pan-Arctic melt pond
fractions and to quantify regional differences in their seasonal evolution. We analyze the ability of the MPD2 products to resolve the temporal and and spatial variability of melt pond processes. We merge our results with sea ice surface roughness data and ice type information from satellite observations to analyze driving mechanisms of melt pond formation and evolution. We further investigate the potential of using melt pond fractions in early summer as an indicator for summer sea ice surface energy budget, including the sea ice minimum extent in September.





## 2 Data Sets

The analysis presented in this work is based on the melt pond fraction product from the MPD2 algorithm developed by Niehaus et al. (2024) which provides melt pond fractions with respect to the sea ice area. The algorithm is applied on Sentinel-3 OLCI data from 2017 to 2023. For the cloud filtering Sentinel-3 SLSTR data is used in addition. For the quantitative analysis of this data product in this study additional steps of filtering are applied for data consistency purposes as presented in Section 2.1. In addition, data sets used for the interpretation and further analysis of the melt pond fraction data are shortly introduced in this section.

### 2.1 Melt Pond Fraction Data Preparation

Since the MPD2 algorithm operates in a wavelength regime where clouds are opaque, cloud pixels must be carefully removed from the data product. The swath data are already filtered for cloud pixels within the MPD2 algorithm itself, according to the criteria defined by Istomina et al. (2010, 2011) and Zege et al. (2015). However, cloud shadows are not captured by this filter but tend to influence the results of the MPD2 algorithm, resulting in an overestimation of melt pond fractions, as illustrated in Figure 1, because the shadowed pixels appear darker. Therefore, an additional filter to remove shadowed pixels is applied to the processed swath data, before the daily maps are compiled. This filter is based on the difference in the ratio of measured top of atmosphere (TOA) reflectances in the blue (490 nm) and red (681 nm) edge wavelengths between observations of shaded and not shaded surfaces:

$$\frac{|\rho(681\,\mathrm{nm}) - \rho(490\,\mathrm{nm})|}{\rho(681\,\mathrm{nm})} > 0.95. \tag{1}$$

If this condition is met, which is usually the case when the TOA reflectance is much higher in the blue range than near the red edge, a pixel is likely to be shadowed and thus removed. In cloud shadows, the observed reflectance originates mainly from scattering in the atmosphere, which is dominated by the Rayleigh scattering and hence inversely proportional to the fourth power of the wavelength. Thus, blue light is scattered more than red light and gives a much higher TOA reflectance than the red light and the condition is fulfilled. Otherwise, in the cloud-free case, the measured signal is dominated by the surface reflectance which shows smaller differences between the two wavelengths. Along the edges of the combined cloud and cloud shadow mask, individual pixels and small patches of retrieved data are additionally removed due to the high risk of undetected disturbances. The filtering is performed using a combination of morphological operations, dilation and erosion (i.e., closing), applied to the combined mask. The number of applied iterations of the dilation and erosion operations $s$, which defines the rigor of the filtering, is chosen as a function of the solar zenith angle (SZA), since the probability of larger cloud shadows increases with a lower sun position (higher SZA) for a constant cloud height:

$$s \propto \tan\left(\frac{SZA \cdot \pi}{180°}\right). \tag{2}$$

After the processing just described, the swath data are assembled into daily maps that are gridded on the NSIDC (2024) grid of 12.5 km resolution. During the gridding process, additional filters are necessary to minimize biases due to insufficient or





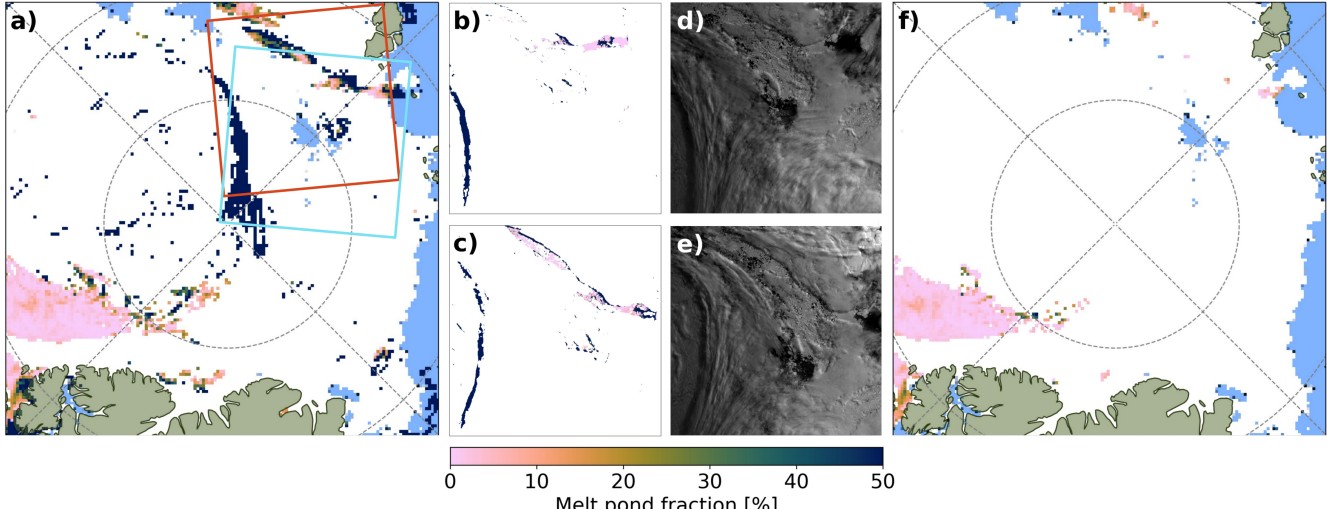

**Figure 1.** Illustration of cloud shadows leading to falsely high melt pond fraction values, for the example of September 7, 2022. a) Excerpt from the daily melt pond fraction map aggregated from the swath data without additional cloud shadow filtering. In the center, there is a large area indicated with high melt pond fractions resulting from cloud shadows. The cyan and red rectangles shows the position of the excerpts from two processed example swaths displayed in b) and c), respectively. Panels d) and e) present the respective raw data at the wavelength of 490 nm in gray scale. They both show an extended cloud shadow near the left edge, which is classified as high melt pond fraction area. The composite of the two presented swaths and data from other overflights accounts for the large, elongated area of very high melt pond fractions in a). f) Excerpt from the daily melt pond fraction map aggregated from the swath data with the additional cloud shadow filter applied to the swath data. The colorbar indicates the melt pond fraction for all maps of processed data.

unevenly distributed data, thereby allowing a reliable discussion of the results. The applied process steps follow the inhomogeneity measure $I$ defined by Sofieva et al. (2014), which is essentially divided into a measure of the data asymmetry $A$ and entropy $E$:

$$I = \frac{1}{2}\left(A + (1 - E)\right). \tag{3}$$

Possible values for $I$, $A$ and $E$ are between 0 and 1. High values of $I$ indicate a strong inhomogeneity of the sampling distribution. The entropy E is 1 for perfectly homogeneously distributed data, decreasing in case of data gaps, and is calculated by partitioning the data into spatial bins or time steps $n$, for spatial or temporal measures, respectively:

$$E = \frac{-1}{log(N)} \sum_i \frac{n(i)}{n_0} log\left(\frac{n(i)}{n_0}\right). \tag{4}$$

Herein, $N$ is the number of bins or time steps, $n(i)$ is the number of observations within the bin or time step $i$, and $n_0$ is the

total sample size. Thus, the entropy describes how much data are missing but it does not take into account the distribution of the remaining data. This is captured by the asymmetry $A$, which is high in case of strongly unbalanced data and 0 if the distribution is perfectly symmetric. It is calculated from the mean spatial or temporal position $\bar{x}$ in a grid cell or time period of





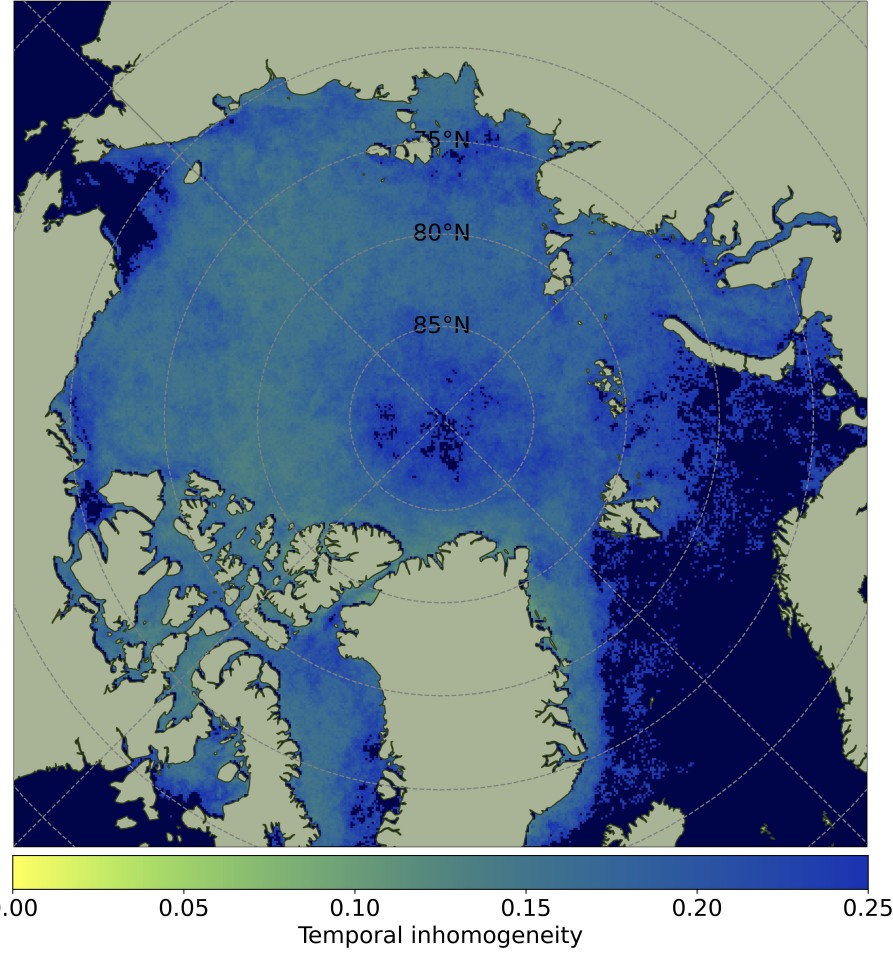

**Figure 2.** Temporal inhomogeneity calculated using Equations 3 to 5 for each grid cell of the north polar stereographic NSIDC 12.5 km resolution grid. The cutoff is set to 0.25 (upper limit) and mainly removes areas that are not covered by sea ice most of the time period. Data that exceed the limit of allowed temporal inhomogeneity or are not within the observed Arctic regions, are colored in dark blue.

width $\Delta x$ and its central position or time $x_0$:

$$A = 2\frac{|\overline{x} - x_0|}{\Delta x}.$$

(5)

First, the temporal inhomogeneity $I_T$ within the 7 year time span is tested pixelwise within the 12.5 km spatial resolution daily maps constructed from the swath data. The temporal inhomogeneity is higher for grid cells with sparse data. This mainly affects areas that are not covered by sea ice for large parts of the observation time period. Additionally, small areas in the high Arctic (above 85 °N) are filtered out because low solar zenith angles at the end of the summer season lead to data gaps at high latitudes first. The filtered areas are indicated by the dark blue color in Figure 2. The threshold of 0.25 was chosen empirically

to preserve the maximum possible amount of data while removing the problematic areas mentioned above. The temporal





inhomogeneity filter is computed from and applied to the daily data to avoid loosing temporal information. Subsequently, the data are aggregated into 7 day composite maps using a running mean centered on the day of interest.

The spatial inhomogeneity is computed separately for each Arctic region (see Figure 4) to ensure that an analysis of average values per region is reasonable. For the same purpose, days are additionally excluded from the time series if less than 20 % of the sea ice area (at the beginning of the observation period) is covered with retrieval data. This removes days with strong

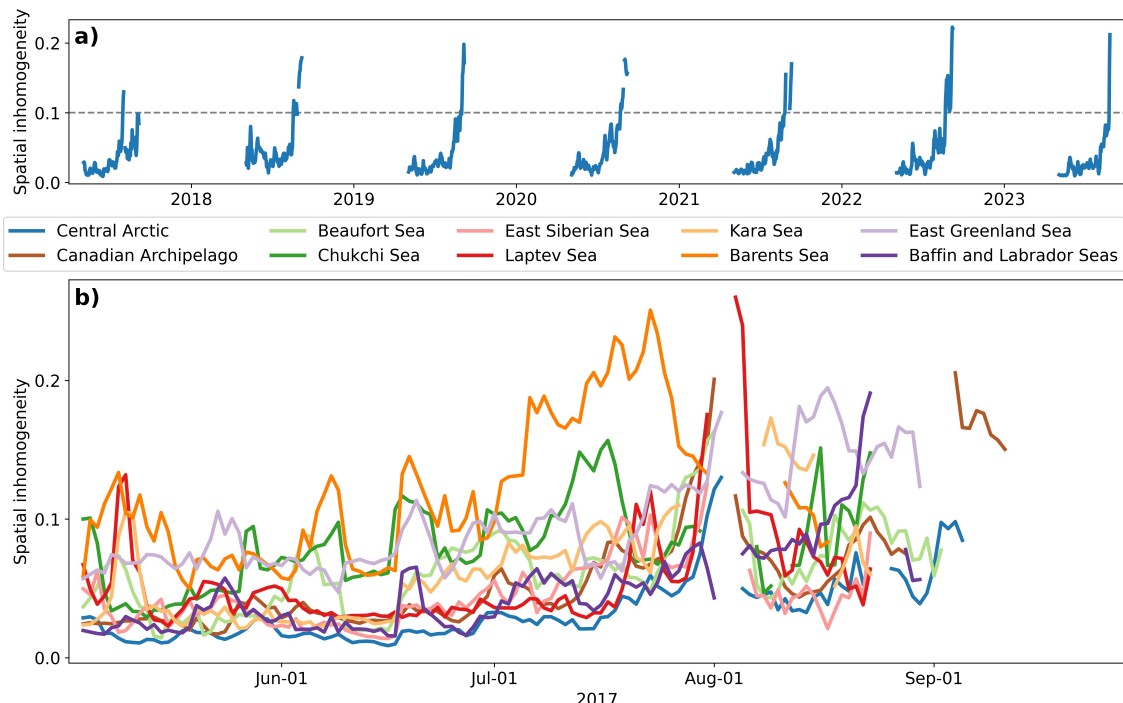

**Figure 3.** Spatial inhomogeneity calculated with Equations 3 to 5 for each day of the observation period. a) Spatial inhomogeneity for the Central Arctic region over the entire 7 year observation period. The dashed horizontal line marks the threshold applied to filter the data. b) More detailed view of the 2017 summer season, for all used Arctic regions.


irregularities in the regional data distribution to ensure that an average represents the entire region. Figure 3 displays the calculated spatial inhomogeneity for the Central Arctic for the entire 7 year period, and for all relevant Arctic regions (see Figure 4) for the summer 2017 season in detail. Gaps in the inhomogeneity measure are due to insufficient data coverage. In general, the spatial inhomogeneity increases during the summer periods. This is in large parts explained by the decreasing sea

ice area and lower sun elevation towards the end of the summer season season, leading to deficient illumination and larger cloud shadows to be filtered. The upper threshold above which days are removed from the seasonal analysis is the median regional spatial inhomogeneity plus two times its standard deviation. This threshold is applied to each region separately and ranges from 0.1 to 0.2.



In this study, any kind of melt water on top of the sea ice is defined as melt pond, even if they are partly melted through the sea ice or have a lateral connection to the ocean.

## 2.2   ERA5 Temperature

The 2-meter air temperatures from the ECMWF fifth-generation reanalysis ERA5 (Hersbach et al., 2020) are used to examine the relationship between melt pond evolution and the temperature forcing. These temperature data are sourced from the ERA5 web page (https://cds.climate.copernicus.eu/cdsapp#!/dataset/reanalysis-era5-single-levels, accessed last on September 30, 2023), featuring a spatial resolution of 0.25°and a temporal resolution of 6 hours.

## 2.3   MOSAiC Campaign Data

For the case study of the melt pond evolution along the drift track of the year-long Arctic research expedition MOSAiC of the research vessel *Polarstern*, in situ observations from that measurement campaign have been used for comparison. An overview of the atmospheric and sea ice observations can be found in Shupe et al. (2022) and Nicolaus et al. (2022), respectively. In this work, particular, air temperatures and precipitation measurements from the meteorological station and the atmospheric flux station (ASFS30) are used. Details on these observations can be found in Cox et al. (2023). Additionally, melt pond fraction observations from in situ transects on the sea ice provided by Webster et al. (2022) are used for comparison.

## 2.4   ASI Sea Ice Concentration

For the assessment of sea ice coverage in the analysis of the regional differences in melt pond evolution, the ASI sea ice concentration (SIC) data product derived from AMSR2 satellite passive microwave observations is used. Details on the retrieval algorithm and data product can be found in Kaleschke et al. (2001) and Spreen et al. (2008). The data has been downloaded from the web page of the University of Bremen (https://data.seaice.uni-bremen.de/amsr2/asi_daygrid_swath/n6250/) with a daily temporal resolution and a 6.25 km spatial resolution.

## 2.5   ICESat-2 Sea Ice Surface Roughness

For the discussion of the correlation between melt pond fractions and the sea ice surface roughness, the pan-Arctic drag coefficient product developed by Mchedlishvili et al. (2023) from ICESat-2 laser altimeter surface elevation data is used. This data set contains monthly averages, resampled to a 25 km polar stereographic (NSIDC, 2024) grid, of several types of drag coefficients as well as obstacle height and obstacle spacing. The obstacle spacing is the average distance between ridges exceeding a height of 20 cm within the grid cells of 25 km edge length. This quantity is applied as a good approximation for the sea ice surface roughness in terms of large-scale deformation relevant to pond evolution, since obstacles with heights greater than 20 cm are considered to be serious barriers to the expansion of surface melt water. The data is available daily for the years 2019 to 2021 on a 25 km on (https://doi.pangaea.de/10.1594/PANGAEA.959728). Details on the data product can be found in Mchedlishvili et al. (2023).





## 2.6 Multiyear Ice Concentration

The multiyear ice (MYI) concentration data product (Melsheimer et al., 2019) developed by Ye et al. (2016a, b) using passive
microwave satellite observations of AMSR2 provides daily pan-Arctic maps of MYI concentration, i.e., the percentage of sea
ice that has survived at least one summer melt season, at a spatial resolution of 6.25 km. To classify sea ice as either MYI
or first-year ice (FYI), thresholds are applied pixel by pixel and the dominant ice type is taken: pixels with $MYI > 70\,\%$ are
classified as MYI and pixels with $MYI < 30\,\%$ are classified as FYI. Because the MYI retrieval was designed for non-melting
conditions, the data are only provided until the beginning of May. Thus, MYI concentrations on May 1 of are used to determine
the areas of MYI and FYI at the beginning of the melt season and relate the melt pond fractions with the sea ice type as presented
in Section 4.4. Details on the retrieval algorithm can be found in Ye et al. (2016a, b) and the data is downloaded from the web
page of the University of Bremen (https://data.seaice.uni-bremen.de/MultiYearIce/).

## 2.7 OSI-SAF Drift Product

In Section 4.4, the low-resolution sea ice drift product of the European Organization for the Exploitation of Meteorological
Satellites (EUMETSAT) Ocean and Sea Ice Satellite Application Facility (OSI-SAF) is used to track the sea ice drift and
thereby map sea ice types. This product offers sea ice motion vector fields over a 48-hour period merged from multiple satellite
drift vectors, employing an optimal interpolation method. The provided drift data come with an uncertainty estimate of 4 to
8 km. This near-real-time product is available daily since 2019, featuring a grid spacing of 62.5 km projected onto the NSIDC
polar stereographic grid.

## 3   Melt Pond Fraction Evolution for Arctic Regions

Understanding the formation and seasonal evolution of melt ponds is fundamental for understanding and modeling the heat
and mass balance of sea ice in the summer season. The here developed pan-Arctic melt pond fraction product offers great
potential to extend existing knowledge from in situ measurements by assessing the large-scale variability and monitoring
regional differences. For this purpose, the Arctic is divided into ten common regions as defined by the National Snow and
Ice Data Center (NSIDC, 2024). These regions are displayed in Figure 4. For each region, the mean values of the melt pond,
open ocean and sea ice fractions are calculated from the aggregated 7-day composites described in Section 2.1. A detailed
overview of the seasonal evolution of the mean values enriched by the probability density distribution of the melt pond fraction
observations is presented in the Figures 5 and 6 for the years 2017 to 2023 and for all regions. Additionally the progression of
sea ice extent in the regions is shown to allow an estimation of the relevance of the melt pond fraction. If there is little sea ice in
a region (e.g., in the East Siberian Sea at the end of the summer season), the impact of high melt pond fractions on the energy
budget is less because this quantity is given relative to the total sea ice area. The sea ice extent is derived from the ASI sea ice
concentration (SIC) data product (Section 2.4) using a typical threshold of $SIC > 15\,\%$ per grid cell. The area defined by this
threshold is also the area where the melt pond fraction retrieval is applied. Thus, the mean sea ice fraction retrieved by MPD2



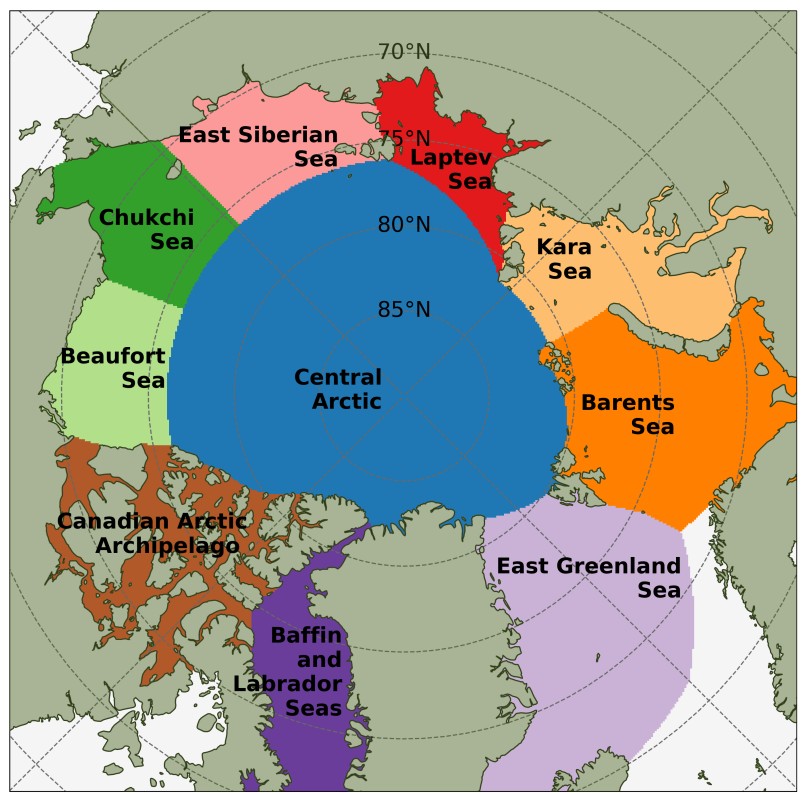

**Figure 4.** Overview of the Arctic regions following the definition by the National Snow and Ice Data Center (NSIDC, 2024).

shown in the figures has a different meaning than the sea ice extent from ASI SIC. While the percentage of sea ice extent refers to the entire ocean in a region, the MPD2 sea ice fraction refers only to the area that has more than 15 % sea ice concentration. It then gives information about the open ocean area within the ice pack, which can be cracks, leads and polynyas.

The largest share and most constant area extent of Arctic sea ice is located in the Central Arctic, as can be seen from the gray sea ice lines in the Figures 5 and 6. The marginal seas of the Arctic Ocean can be classified in three groups based on

the seasonal cycle of sea ice extent. The first group includes the the Barents Sea, the East Greenland Sea and the Baffin and Labrador Seas: these regions are never fully covered by sea ice and typically become completely ice-free by the end of the summer (dashed gray lines). The second group is the opposite and includes only the Canadian Arctic Archipelago and the Beaufort Sea: these regions are completely ice-covered in the winter and retain some sea ice during the sea ice minimum extent in late summer. However, the late summer sea ice cover in the Beaufort Sea is highly variable and there are exceptions when

this region also becomes ice-free. It would then fall into the third group that is otherwise contains the Chukchi Sea, the East Siberian Sea, the Laptev Sea, and the Kara Sea: all these Arctic marginal seas are close to 100 % ice covered at the beginning of the melt season and tend to become ice-free at the end of that season. Further characteristics to subdivide the Arctic marginal seas are the impact of sea ice drift and the presence of landfast ice. The latter is restricted in many regions to the beginning





of the melt season, as landfast ice means that the sea ice is attached to the land masses, which are generally located at lower

latitudes and thus experience earlier melt. A special case is the Canadian Arctic Archipelago with its large number of islands, where more than 80 % of the sea ice is landfast ice at the beginning of the melt season (Galley et al., 2012). Other Arctic regions with larger amounts of landfast ice (areas of more than $10^5 \, \text{km}^2$) at the beginning of summer are the East Siberian, the Kara and the Laptev Seas. The Barents and Chukchi Seas, on the opposite, contain the least amount of landfast sea ice (Yu et al., 2014; Dammann et al., 2019).

## 3.1   Central Arctic

The Central Arctic region shows the smallest and latest increase in melt pond fraction per year, with median melt pond fractions not exceeding 30 %. This is consistent with the high stability of sea ice (extent). In all years, the overall melt pond fraction remains close to 0 % until late June, with a stronger increase towards early July. This can be largely explained by the extremely high latitudes and consequently cold average temperatures. Bimodal distributions of the melt pond fraction are a common

feature, with some areas in the central Arctic region having no melt ponds at all and some other areas having high melt pond fractions up to 50 %. In some years, e.g., 2020 and 2023, there are periods of temporary melt pond presence throughout the entire Central Arctic with spatially averaged values between 20 % and 50 %. This is in agreement with observations by Istomina et al. (2015b), Istomina et al. (2023) and Rösel and Kaleschke (2011).

## 3.2   Canadian Arctic Archipelago

Melt pond fractions in the Canadian Arctic Archipelago show a sudden increase in mid-June from 0 % to a median of 40 %, usually in less than two weeks, leading to maximum melt pond fractions in early July. Meltwater can spread quickly across the sea ice surface if temperatures are high enough, because much of the Canadian Arctic Archipelago is covered by landfast ice, which tends to be little deformed and rather flat. Peak melt pond fractions are typically between 40 % and 50 %, which is consistent with observations by Howell et al. (2020). The sea ice fraction within the ice-covered area, which is an estimate of

how compact the sea ice is, remains at 100 % until early July, when the peak melt pond fractions are reached. Subsequently, sea ice compactness decreases slightly, coinciding with a decrease in melt pond fraction, as melt ponds begin to melt through the ice (Istomina et al., 2023). This is also in line with the onset of drainage dated to the end of June for landfast ice in the Canadian Arctic Archipelago by Tanaka (2020) using AMSR-E passive microwave data. They also estimate the duration of the drainage period to be about 15±5 days, which is consistent with the observation that the melt pond fraction tends to increase

again in mid-July due to enhanced lateral melt.

An exceptional year is 2023, when melt pond fractions took more time to increase, but then did not drop in between, reaching more than 50 % median melt pond fraction in late July. During the periods of maximum melt pond fraction, the probability density distribution usually shows no areas of no melt ponds at all, but a broad distribution around the median with local values up to 80 %.



**Figure 5.** Seasonal evolution of melt pond fraction for seven years of Sentinel-3 observations, differentiated by Arctic regions as shown in Figure4. The remaining regions are shown in Figure 6. The colorbar indicates the probability density of the melt pond fraction distribution and is valid for all subfigures. The median melt pond fraction is considered as fraction of sea ice area. Data gaps in the MPD2 output are due to insufficient data coverage.



**Figure 6.** Continuation of Figure 5. Seasonal evolution of melt pond fraction for seven years of Sentinel-3 observations, differentiated by Arctic regions as shown in Figure 4. The colorbar indicates the probability density of the melt pond fraction distribution and is valid for all subfigures. The median melt pond fraction is considered as fraction of sea ice area. Data gaps in the MPD2 output are due to insufficient data coverage.





### 3.3 Beaufort Sea


The Beaufort Sea is located at similar latitudes and directly adjacent to the Canadian Arctic Archipelago, but contains very little landfast ice. In contrast, it is characterized by high sea ice drift activity caused by the Beaufort Gyre (Reimnitz et al., 1994). Thus, the seasonal melt pond evolution is much more variable compared to that in the Canadian Arctic Archipelago, despite the similar timing of the first strong increase of melt pond fraction in late June. Later in the melt season no coherent

pattern is visible for the years monitored.

### 3.4 Chukchi Sea, East Siberian Sea, Laptev Sea and Kara Sea

These Arctic marginal seas exhibit the strongest seasonal cycle in sea ice extent. They go from near full coverage in spring to almost completely ice-free by the end of summer, as indicated by the dashed gray lines in Figures 5 and 6 . Overall melt pond formation is highly variable, with melt pond fractions increasing as early as late May (e.g., Chukchi and East Siberian Seas

in 2020 and Laptev and Kara Seas in 2023) or as late as July (e.g., Kara Sea in 2017). Most commonly, melt pond fractions increase in late June. Overall, median melt pond fractions show high interannual and seasonal variability. Among these four regions, the highest melt pond fractions are observed in the Laptev Sea, with peak median values up to 45 %, which typically occur in the first half of the melt season, most likely when landfast ice is largely ponded and not yet melted. The Laptev Sea typically has one of the largest landfast ice areas of the four marginal seas (Dammann et al., 2019), which is consistent with

having the highest melt pond fractions.

### 3.5 Barents Sea, East Greenland Sea and Baffin and Labrador Seas

The Barents Sea is close to ice-free at the earliest (in July), and it rarely develops mean melt pond fractions of more than 25 %. This suggests that the sea ice melts more from below than from the surface, which may be due to increased heat transport into the Barents Sea from the North Atlantic Ocean current, referred to as *Atlantification*. (Årthun et al., 2012; Lundesgaard

et al., 2022; Polyakov et al., 2023). The other two regions in this group, with less than 50 % sea ice at the beginning of the melt season, the East Greenland Sea and the Baffin and Labrador Seas, have higher median melt pond fractions and also higher extreme values, which is likely related to the extension of sea ice to lower latitudes in these regions.

In the East Greenland Sea, both seasonal and interannual variability are high. This variability is most likely driven by the strong sea ice drift and export through the Fram Strait initiated by the transpolar drift pattern (Spreen et al., 2020; Smedsrud

et al., 2017). High sea ice export rates can lead to sudden declines in the melt pond fraction, which can vary greatly from year to year. In the Baffin and Labrador Seas, in contrast, the seasonal evolution is much more similar between years. Typically, melt ponds begin to form in mid-June, and median melt pond fractions rise to about 40 % in July, with peaks much higher and a bimodal distribution.

Hereinafter we focus on the following regions, representing different types of sea ice conditions:

– **Central Arctic:** has the most seasonally stable sea ice extent and contains most of the existing multiyear ice (MYI).





- **Canadian Arctic Archipelago:** contains the largest area of landfast ice and is never completely ice-free.

- **Beaufort Sea:** sea ice is characterized by high drift activity due to the Beaufort Gyre, almost no landfast ice.

- **Kara Sea:** combination of seasonal landfast and drift ice

Figure 7 compares the seasonal evolution of melt pond fraction for these 4 regions and all 7 years. The underlying Sentinel-3
data are the same as in the Figures 5 and 6. In addition, the time period of average melting temperatures per year and region is highlighted using ERA5 reanalysis data, and the exceedance of certain melt pond fraction values is marked to allow a better comparison between years. The same information for the other six subregions are shown in Figure A1 in the appendix but not further discussed here.

For the Central Arctic and the Canadian Arctic Archipelago, there is no obvious change in the duration of the melt season
between years. On average, melting temperatures are first reached around the beginning of June and end in late August or the first half of September for the Central Arctic and the Canadian Arctic Archipelago, respectively. In the Beaufort and Kara Seas, there is a slight trend towards longer melting periods: in the later years presented, melting conditions start earlier, as early as May, and temperatures do not fall below 0 °C before mid-September. The pond formation, defined by average melt pond fractions exceeding the MPD2 algorithm uncertainty of 8 %, is quite consistent between years except for the Kara Sea,
where the timing of pond formation varies between mid-May and late June. In the other regions, pond formation typically occurs within a week between mid-June in the Beaufort Sea and early July in the central Arctic. Interestingly, there is no clear correlation between the timing of melting temperatures and the formation of the first melt ponds. The time between these events can vary from a few days to more than a month. This could be due to averaging over large regions, where small areas could already experience pond formation that would not be visible in the average, but could also indicate long periods of temperatures
meandering around 0 °C.

Higher melt pond fractions are generally observed in the Canadian Arctic Archipelago and the Beaufort Sea with prominent maxima in 2023. In the Canadian Arctic Archipelago, the pond evolution initially appears relatively smooth and uniform, which may be related to the easier expansion of melt ponds on flat landfast ice. However, the timing of maximum pond coverage varies between early July and late August, which is also the case in the Beaufort Sea, demonstrating the increasing
complexity towards the end of the melt season.

The Central Arctic shows the smoothest seasonal evolution, while the Kara Sea is the most variable. As discussed above and shown in the Figures 5 and 6, this can largely be attributed to the changes in sea ice extent. In the Central Arctic, there is generally more time between the onset of melting and the formation of ponds. However, once ponds have formed, their evolution is fast, and 75 % of the maximum melt pond fractions are usually reached within less than 10 days, and the maximum
is often reached shortly thereafter. In the Kara Sea, more time elapses between pond formation and maximum melt pond fractions, with an overall high variability. An extreme case is observed in 2023, where pond formation starts earliest and maximum pond fraction is reached latest.

Table 1 summarizes the timing and extend of melt pond formation and evolution for the four focus regions.



**Figure 7.** Seasonal evolution of melt pond fraction for 7 years for the Central Arctic, Canadian Arctic Archipelago, Beaufort Sea, and Kara Sea. The color indicates the daily averaged melt pond fraction with respect to the sea ice area per region according to the color scale. Gray areas represent data gaps. The vertical black lines mark the beginning and end of the period when regionally averaged temperatures are above 0 °C. If the end is not marked, it is beyond the time period shown. The light blue colored triangles indicate the pond formation, defined by melt pond fractions exceeding the uncertainty of the MPD2 product of 8 %. The red circles mark the time of the regional maximum melt pond fraction per year, and the dark triangles mark the day when 75 % of the maximum is exceeded.

## 4  Mechanisms of Melt Pond Evolution

A vast variety of aspects can influence the initial formation of melt ponds, their seasonal evolution and drainage, and thus the overall spatial and temporal heterogeneity. Here, some of these variables are collected from other data sources and contextualized with the MPD2 melt pond fraction and open ocean fraction products. Correlations are investigated at different spatial scales. First, a special case from the MOSAiC expedition (Nicolaus et al., 2022; Shupe et al., 2022) is examined at the 1.2 km



**Table 1.** Summary of basic information on melt pond evolution for the four regions of focus.

|  | Central Arctic | Canadian Arctic Archipelago | Beaufort Sea | Kara Sea |
|---|---|---|---|---|
| Peak median pond fractions | 15 - 30 % | 40 - 50 % | 25 - 50 % | 25 - 45 % |
| Timing of maximum pond fractions | early July - early August | late June - late August | mid-June - mid-August | mid-June - mid-August |
| Timing of melt pond formation | late June - early July | early June - mid-June | mid June | late May - late June |
| Seasonal variability | low | medium | high | medium - high |
| Interannual variability | medium - low | medium - low | medium - high | medium |

spatial resolution of the Sentinel-3 OLCI swath data. Subsequently, regional and pan-Arctic studies are presented at the 12.5 km
spatial resolution of the final melt pond fraction maps.

## 4.1 Focus: MOSAiC Drift in 2020

During the MOSAiC campaign atmospheric conditions and weather events have been continuously recorded in addition to
detailed surface observations including albedo measurements. This documentation gives indication of an early and short-lived
melt event near the Polarstern research vessel in May 2020 (Webster et al., 2022), which was at that time located north of
Svalbard in the Central Arctic region. The regional averages shown in Figures 5, 6 and 7 cannot capture such small scale
events, but the probability density distribution in Figure 5 shows some occurrences of increased melt pond fraction for late
May 2020 in the Central Arctic. The satellite swath data of MPD2 have a footprint of 1.2 km and can be used to study local
changes in melt pond fraction that are not well observed in the aggregated 12.5 km resolution product. Along the MOSAiC drift
track between May and August 2020, the processed swath data of all matching overflights in cloud-free situations is collected
and melt pond fractions are averaged for an area of 20 km radius around the current position of the MOSAiC ice floe. These data
are presented in the lower panel of Figure 8. The dark gray shaded area highlights the short melting period in May mentioned
above. During this period, rainfall and widespread ponded surfaces have been reported by Webster et al. (2022), which is
consistent with the average melt pond fractions between 20 % and 26 % observed in three cloud-free Sentinel-3 overflights on
May 29.
The 2 m air temperature rose above 0 °C for the first time in this summer season on May 26, which was well observed by the
in-situ observations as well as the reanalysis data, shown in the middle panel of Figure 8. Thus, the precipitation measured by
the in situ instruments is rain rather than snowfall, as reported by Webster et al. (2022), leading to an enhanced watering of the
sea ice surface and high melt pond fractions, as shown in the lower panel of Figure 8. This observation is consistent with the





**Figure 8.** Temporal evolution of melt pond fractions on sea ice and atmospheric conditions along the drift track of the MOSAiC floe in summer 2020. a) Precipitation rates from ERA5 (dark blue dots) and MOSAiC in situ measurements (light blue) for the same area and from the same stations as for the temperature. b) Temperatures at 2 m height averaged over the same area as the melt pond fractions, from ERA5 (dark red dots) and from the met station and the ASFS30 deployed on the sea ice during the MOSAiC campaign (Cox et al., 2023) (light red). c) Average melt pond fractions of the 20 km radius area around the MOSAiC floe position, retrieved with MPD2 from Sentinel-3 satellite overflights (dark green dots) and in situ observations of melt pond fractions by Webster et al. (2022) (light green). The dark shaded area highlights the melt period reported by Webster et al. (2022), the light shaded area highlights another case of a short-lived increase in the melt pond fraction just before the start of the major melt pond period.

local observation of a melt pond and corresponding low albedo at one of the radiation stations deployed in the vicinity of the

MOSAiC floe, reported by Tao et al. (2024). However, the amount of precipitation is not well captured by the reanalysis data,





which is a general problem. According to the in situ measurements, temperatures dropped again between May 29 and 30, and the precipitation turned to snow. Thus, the melt ponds froze over and were covered by snow, which significantly reduced the observed melt pond fraction on May 30. The temperature drop was not well observed in the reanalysis data, where temperatures seem to remain slightly above 0 °C most of the time afterwards, which is a commonly observed bias over sea ice (Wang et al.,
2019; Tian et al., 2024).

A second case of a short-lived melt pond fraction increase was observed on June 18 (light gray highlighted area in Figure 8). This increase was not mentioned by Webster et al. (2022), but fits well with the albedo decrease between June 10 and 18 at one of the radiation station sites reported by Tao et al. (2024), which also demonstrates the high small-scale melt pond variability. The short duration of the increase in the melt pond fraction may be explained by the snowfall observed by the in situ weather
station on June 20 at −2 °C temperatures, which is not identified in the ERA5 reanalysis data. MPD2 melt pond fractions were about 5 % in the monitored area on June 22 and 23. Subsequently, the major melt pond fraction period of the MOSAiC drift began, and according to satellite observations within the 20 km radius, melt pond fractions rose up to of 40 % on August 7, with minor fluctuations in between. This is significantly higher than the melt pond fractions reported by Webster et al. (2022) on their transect route on the MOSAiC central observatory (CO), but also covers a much larger area. Krumpen et al. (2021) and
Niehaus et al. (2023) concluded in their studies that the sea ice conditions of the MOSAiC CO was representative for the wider surroundings except for the initial melt pond formation, which occurred earlier in the CO than in the surrounding area. Melt pond fractions retrieved from 10 m resolution Sentinel-2 observations along the MOSAiC drift track presented in Niehaus et al. (2023) show values in between the high resolution in situ and the low resolution MPD2 data, demonstrating the importance of the observational scale for the data interpretation. In addition, a bias of −5 % for the in situ observations on the CO has been
reported, which fits with overall higher melt pond fractions retrieved from satellite. There are no cloud-free Sentinel-3 passes later than August 7 in this summer season.

Overall, this case study demonstrates the ability of the newly developed MPD2 melt pond fraction product to monitor the high temporal variability of melt pond evolution. However, this study also demonstrates the deficiency of reanalysis data to explain the driving mechanisms of melt pond formation and evolution, partly due to the insufficient spatial resolution of ERA5
reanalysis data. The discrepancies between in situ and ERA5 precipitation lead to the conclusion that this quantity cannot be used on pan-Arctic scales to explain short-term changes in the melt pond fraction. The comparison of in situ and reanalysis temperatures shows better agreement with a common bias towards higher temperatures by ERA5 (Wang et al., 2019; Tian et al., 2024).

## 4.2 Focus: Kara Sea in 2023

Here we investigate the causes for the striking early increase of melt pond fraction in the Kara Sea in late May 2023 (Figures 6 and 7). Figure 9 shows the median and probability density distribution of melt pond fraction in the Kara Sea region for May and June 2023, and the maps of the melt pond fraction, open ocean fraction, and 2 m air temperature in this region for some selected days. On May 12, the entire region is covered by sea ice and melt ponds have not yet formed, as temperatures are generally below 0 °C. Only near the Kara Strait, the melt pond fractions are already slightly increased and the open ocean



fractions more strongly increased, which does not significantly change the regional median value. By May 19, the melt pond fraction variability increases and the median value departs from 0 %.

This change was consistent with increasing air temperatures and was mostly confined to latitudes below 73 °N or near the coast. In the following week, temperatures rose above the melting point throughout the region, reaching above 5 °C in the southern part, leading to strong surface melt that peaks on May 26 to 27. The melt pond fraction map shown for May 26

indicates melt pond fractions between 20 % and 40 % up to 76 °N. At higher latitudes, temperatures remained closer to freezing and melt pond fractions near 0 %. Open-ocean fractions increased in the southernmost part of the region and in coastal areas, where small areas were already close to ice-free (light blue areas in Figure 9) according to the ASI sea ice concentration product of Spreen et al. (2008). Subsequently, the melt pond fractions slowly decreased, but the open ocean fractions increased. It is likely that high air temperatures and sea ice motion, which tends to be higher in the southern part of the Kara Sea (see

Figure A2), cause the sea ice cover to be highly fractured, allowing easy drainage of melt ponds. Especially in the Gulf of Ob, melt pond fractions are again close to 0 %, while the open ocean fraction is high on June 3. This is also the only part of the Kara Sea where the temperature remained on a high level, while large parts again experienced temperatures close to 0 °C, as shown for June 3 and 11. Overall, the melt pond fraction variability was strongly reduced by June 11, and the median value was very low. This can be explained by the fact that larger areas that previously had high melt pond fractions are now sea ice-free, while

melt pond fractions at higher latitudes are still low. The complete break-up of sea ice was most evident in areas that previously had high melt pond and open ocean fractions. In addition, Figure A2 indicates slightly increased sea ice drift activity in the Kara Sea at latitudes below 76 °N, which promoted the break-up of the already thinned sea ice (Rabault et al., 2023). After June 15, the main melt season began in the Kara Sea. By June 22, melt ponds had formed almost everywhere in the region.



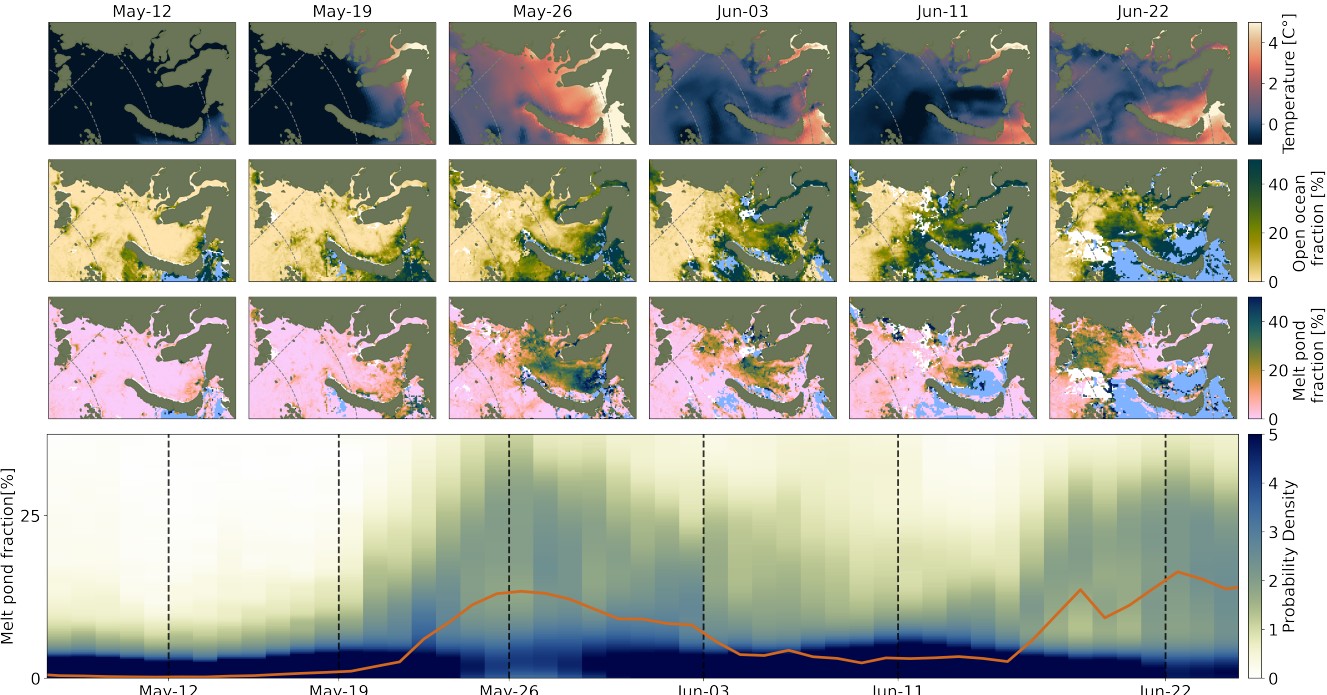

**Figure 9.** Case study of melt pond evolution on sea ice in the Kara Sea in spring to summer transition in 2023. Bottom panel: median melt pond fraction (orange line) and probability density distribution of the retrieved melt pond fraction in the Kara Sea region in May and June 2023. Vertical dashed lines mark the dates of the maps in the rows above. Third row: 7-day composite melt pond fraction maps of the Kara Sea region, retrieved with MPD2 and displayed for the days marked in the lower panel. Second row: 7-day composite open ocean fraction maps of the Kara Sea, retrieved with MPD2 for the same days. Top row: 2 m air temperatures from ERA5 for the same region and days. The color scales on the right side are unified per row. Light blue areas in the second and third rows indicate sea ice free areas according to the ASI sea ice concentration.

### 4.3 Sea Ice Surface Topography and Latitude

To study the influence of sea ice surface topography on melt pond formation and evolution, the obstacle spacing between ridges as provided in pan-Arctic drag coefficient product by Mchedlishvili et al. (2023) and described in Section 2.5 is used. This quantity is assumed to be a good proxy for surface roughness in terms of large-scale deformation relevant to pond evolution, since obstacles with heights greater than 20 cm are considered to be serious barriers to the expansion of surface melt water. In the following, sea ice with small obstacle spacing is also referred to as rough or deformed ice, and sea ice with large

obstacle spacing is also referred to as flat or level ice. The impact of the sea ice surface topography, i.e., the obstacle spacing, is differentiated between latitudinal bands to eliminate a strong influence of temperature on the analysis. Moreover, the impact of the obstacle spacing on the probability density distribution of melt pond observations is monitored in weekly steps over the summer season from late May to early August, as shown in Figure 10. At the end of May, there was little variation in melt





**Figure 10.** Latitude and surface roughness dependence of melt pond fraction distributions on the Arctic sea ice for the years 2019-2021. The rows represent the evolution during the summer season divided into weeks, the colors of the lines indicate the roughness classes based on the obstacle spacing from the data developed by Mchedlishvili et al. (2023). The vertical dashed lines mark the mean values. The distributions are shown for a limit of at least 1000 valid data points. The colors on the right side of each panel indicate the mean air temperature per latitude band and time period.





pond fraction observations between sea ice of different surface roughness (Figure 10). In the following for simplicity we look
at latitudinal bands of 5°to account for temperature changes and discriminate the marginal ice zone from the inner ice pack
(we are aware that actual atmospheric and ocean variability does not strictly follow latitudes). At all latitudes, the probability
distribution was highest at 0 % melt pond fraction, with a broader extension and higher mean values the lower the latitude.
As time progressed and average temperatures exceeded 0 °C (reddish color stripe right of plots) the distributions continued to
broaden with average melt pond fractions of about 20 % at latitudes below 75 °N, of about 8 % at latitudes between 75 °N and
80 °N, of 5 % between 80 °N and 85 °N, and still close to 0 % melt pond fraction above 85 °N in the second week of June (third
row from left to right in Figure 10).

Only one week later, June week 3, the average melt pond fractions at the lowest latitudes were close to 40 %, and also
in the other latitude bands melt pond fractions increased strongly. This latitude dependence is consistent with the fact that
temperatures rose less at higher latitudes, as indicated by the colored stripes on the right side of each panel in Figure 10. Until
then, the melt pond fraction distributions were generally broader, with higher mean values for flat ice compared to deformed
sea ice, as indicated by the line colors in Figure 10. This can be explained by the physical limitation of melt water expansion by
ridges, resulting in deeper melt ponds with lower surface coverage (Wang et al., 2018). For latitudes below 75 °N the peak at
0 % melt pond fraction completely disappeared, and the entire sea ice was partially covered by melt ponds. The most common
coverage for strongly deformed sea ice is between 10 % and 30 %, for level ice this range is shifted up by 10 %. With increasing
latitude, the peak of the distribution shifts to lower values, close to 0 %.

By the end of June (week 4), the situation diverged: despite persistently high temperatures, melt pond fractions started to
decrease at low latitudes, while they still increased at higher latitudes. This can be explained by the complete melt of sea ice in
the marginal ice zone, where melt pond fractions were previously high, as discussed for the Kara Sea in Section 4.2. Another
possible explanation is a temporarily increase in sea ice permeability which enhances drainage through the sea ice (Perovich
et al., 2002b). This process, however, can not be well identified and attributed in these data of reduced spatial resolution.
In addition, the trend for rough sea ice to have lower melt pond fractions decreased and even reversed in July. At latitudes
below 75 °N the impact of surface roughness was negligibly small for the remainder of the melt season, but at higher latitudes,
lower melt pond fractions were consistently observed on flat ice compared to higher fractions on deformed ice. This effect was
strongest in the first half of July and at high latitudes. Here, for the case of high surface roughness, a strong maximum of the
distribution developed at pond fractions between 20 % and 25 %, which is a typical average value observed in the advanced
melt season on deformed ice (Fetterer and Untersteiner, 1998; Eicken et al., 2002; Tschudi et al., 2008; Webster et al., 2015).
The specific shape of the peak weakenes with decreasing surface roughness, especially in the first half of July, when the most
likely value of melt pond fraction on level ice was 0 %. Usually, deformed sea ice is also thicker than level ice (von Albedyll
et al., 2020), which is more vulnerable to ice dynamics (Shokr and Sinha, 2023) and prone to drainage through cracks or by
ice melting through. Later in the melt season, the distributions of different surface roughness moved closer together. However,
flat ice tended to have a broader distribution of likely melt pond fractions.

Overall, an influence of surface topography on the evolution of melt pond fractions on Arctic sea ice can be clearly observed
with the presented data set. Taking into account the influence of the slight latitudinal temperature bias, it is still noticeable that





ice deformation limited melt pond coverage early in the melt season, which caused flat, level ice to have a higher melt pond
fraction. Later in the season, flat sea ice showed lower mean melt pond fractions, which likely is related to the on average lower
thickness of the level ice which thus earlier melted through and drained the melt ponds.

## 4.4  Sea Ice Type

The physical properties of the two ice age related sea ice classes, multiyear ice (MYI) and first-year ice (FYI) differ in many
respects (Maykut et al., 1992; Eicken et al., 2002). Apparent macroscopic differences concern sea ice thickness and surface
topography (Johnston, 2017), but sea ice structure, its salinity and porosity, also evolve over time (Untersteiner, 1968). In order
to investigate the impact of the ice type retrieved from (Melsheimer et al., 2019) as described in Section 4.4 on the seasonal
evolution of melt pond coverage, we perform a case study for 2023. During summer no information on ice types is available.
Therefore the daily retrieved melt pond fractions from MPD2 are associated with the sea ice type at the beginning of the melt
season. For this purpose, the OSI-SAF drift data used in the MPD2 algorithm itself and described in Section 2.7 are applied.
Figure 11 shows the results for the Central Arctic in the year 2023. The study is limited to the Central Arctic region (shown in
Figure 4) to reduce the effect of ice completely melting away during summer, which would lead to an inconsistent integration
of ice area and, thus, would complicate the interpretation. However, the actual sea ice included in the analysis may still change
between days because of sea ice drifting out of the region of interest.

   In contrast to the observations summarized by Polashenski et al. (2012), where melt ponds tended to form first and more
extensively on FYI, this study shows earlier melt pond formation on MYI. In mid-June, there was a first peak in melt pond
fractions on MYI, which decreased again towards early July. Meanwhile, melt pond fractions on FYI started to increase later
but then more drastically, far exceeding pond fractions on MYI. The maximum of median melt pond fraction on FYI in the
entire melt season was 34 %, observed on July 10. Such high median melt pond fractions were never observed for the on
average more deformed MYI. This highlights again the influence of surface roughness on melt pond fraction discussed in
the last section. After July 10, melt pond fraction on FYI slowly decreased until early August, when a short-lived minimum
of about 10 % was reached. Later in August, melt pond fractions on FYI increased again to a median value of 25 %. While
the evolution on FYI looks smooth, MYI showed significantly higher variability. Here, the melt pond fractions rose and fell
several times within the season, with two major peaks in mid-July and around August 10. These fluctuations cannot be clearly
attributed. At the time of the second peak, melt pond fractions on FYI were significantly lower. Overall, the maximum median
melt pond fractions of the season were not significantly different between the two ice types (34 % on FYI and 37 % on MYI).
Thus, in the year 2023, the ice types cannot be separated by maximum pond fractions on a specific day alone, as generally
suggested by Eicken et al. (2004). This discrepancy may be due to the use of completely different types of observations at
different spatial resolutions. Eicken et al. (2004) analyzed the differences between FYI and MYI properties and melt pond
fractions using spatially restricted in-situ observations during two measurement campaigns in the Chukchi Sea. This study
considers averages over much larger regions but at lower spatial resolution. At the 12.5 km resolution grid used here, melt pond
fractions as high as 60 %, as reported for FYI by, e.g., Eicken et al. (2004) and Scharien and Yackel (2005), are unlikely to
be observed, while they are quite possible for in situ transects or small scale airborne observations. However, if the melt pond





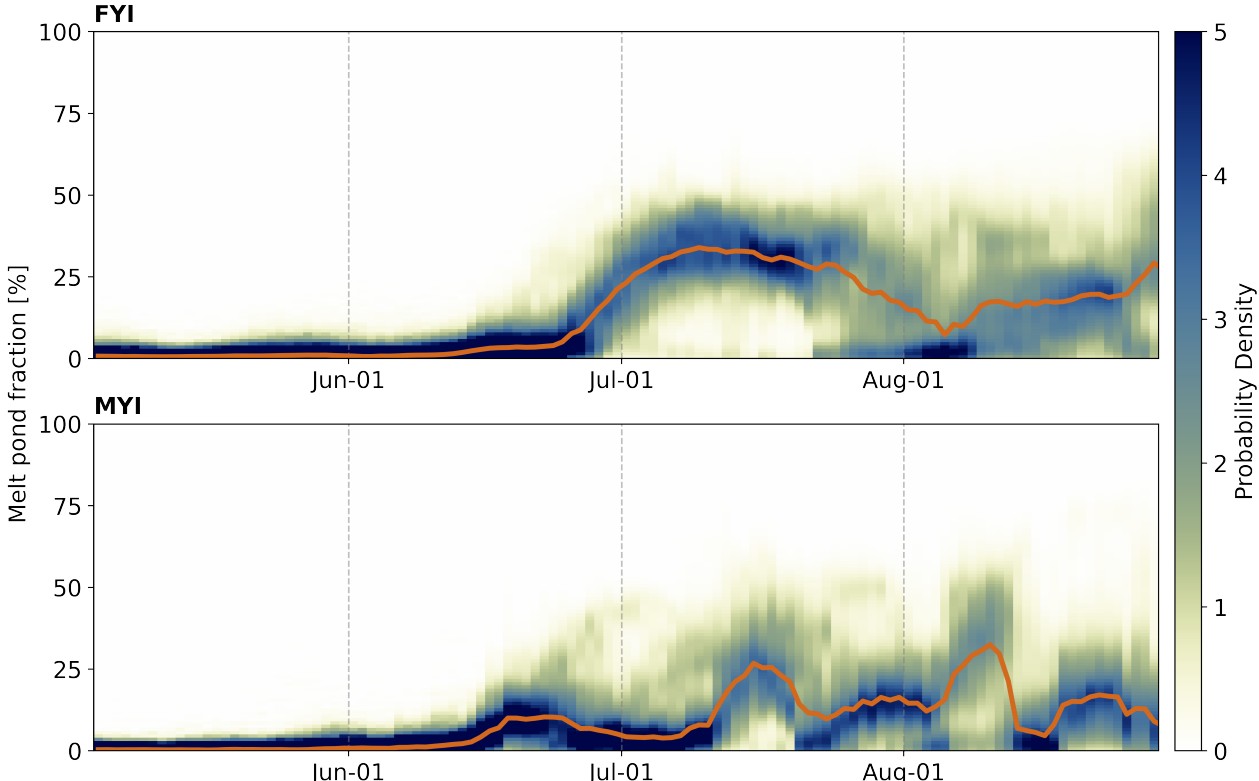

**Figure 11.** Seasonal evolution of melt pond fraction with respect to the sea ice area in the Central Arctic in 2023, differentiated by ice type. The upper panel presents the evolution for first-year (FYI) and the lower panel for multiyear ice (MYI). The color bar indicates the probability density of the melt pond fraction distribution, the orange line marks the median value of melt pond fraction.

fractions shown in Figure 11 are integrated over the whole season, FYI indeed showed a higher averaged value (19 %) than the MYI (10 %) as would be expected due the lower average surface roughness of FYI (see also last Section 4.3).

Instead of maximum values, the seasonal variability of melt pond fraction was the most striking difference between the ice types. Melt pond fractions on MYI appeared to be much more variable, which could be attributed to the more complex surface structure with more ridges and depressions (von Albedyll et al., 2020). As melt water accumulates in depressions, this could explain the earlier increase in melt pond fraction on MYI compared to FYI. However, this was generally not observed in Section 4.3.

The melt pond fraction comparison study by Lee et al. (2024) shows differences between monthly averaged melt pond fractions on MYI and FYI of 2 % to 17 % depending on the data product and month, while we only observe differences of 3 %, 4 %, 10 % and 3 % for the months May to August, respectively. Overall, they report higher variability on FYI at the same spatial resolution as used here. However, as also pointed out by Polashenski et al. (2012), high interannual variability must be





taken into account, which makes it impossible to draw firm conclusions from this study, which currently only covers the year
2023.

# 5   Implication of Melt Pond Fractions on the Annual Sea Ice Minimum

The annual minimum extent of Arctic sea ice usually occurs in mid-September and is a measure of great importance for monitoring climate change, making future predictions, and improving climate models, as well as being of interest to the shipping industry. Due to the sea ice albedo feedback, a positive anomaly in melt pond fraction would lead to an imbalance
of the surface surface energy budget promoting more sea ice melt and thinner ice which possibly lead to a low (regional) sea ice minimum area. And some models indeed show this effect. Using simulated melt pond fraction data, Schröder et al. (2014) show that the September sea ice extent is correlated with melt pond fractions in May by a coefficient of $-0.8$. Liu et al. (2015) use melt pond fractions retrieved from MODIS observations and report a similarly high correlation when they accumulate melt pond coverage not only for May but until the end of June, looking at data between 2000 and 2011. Using only May data, they
find no correlation. Both studies look at the pan-Arctic region as a whole and do not distinguish between smaller sub-regions. We here analyze the effect of May and June melt pond fractions for the pan-Arctic region as well as for the subregions discussed in Section 3.

The dates of the annual sea ice minimum are taken from the NSIDC (2023) website. The minimum sea ice extent is then averaged over the entire week prior to the annual minimum to reduce the effect of daily variability due to retrieval uncertainties
and short-term changes. Correlation coefficients are calculated for the 7-year vector (of Sentinel-3 observations) of minimum sea ice extent and vectors of melt pond coverage or open ocean area accumulated for (a) May only and (b) for combined May and June. In addition, the correlation with air temperature averaged for either one or two weeks before the sea ice minimum date is computed. The resulting coefficients for melt pond area and air temperature are shown in Figure 12, those for open ocean area are not shown as they did not show a consistent effect.

Overall, there appears to be either no correlation or rather an anticorrelation between the extent of the September sea ice minimum and the accumulated melt pond covered area of the early melt season, depending on the region considered. The same is true for air temperatures in the period just before the sea ice minimum. In both cases, considering longer periods leads to a higher anticorrelation (dark colors in Figure 12). In the marginal seas, the May to June melt pond coverage has little or no effect on the extent of the sea ice minimum. This can be explained by the fact that most of these regions are sea ice-free in
September in any case. In addition, the sea ice extent is very sensitive to winds and ocean currents. In the Kara Sea, the air temperature just before the minimum seems to have a strong influence on the minimum extent. It seems to be the decisive factor that determines whether the already thin and unstable ice melts completely or not. Air temperatures also play a role in the Central Arctic, but here the effect of early summer melt pond coverage shows the strongest anticorrelation. The presence of early melt ponds appears to alter the thermodynamic properties of the sea ice, leading to increased melting and easier break-up
later in the season. As the pan-Arctic region is dominated by its largest region, the Central Arctic, it can be concluded that the early summer melt pond fraction has a significant impact on the extent of the September sea ice minimum. It is not the only



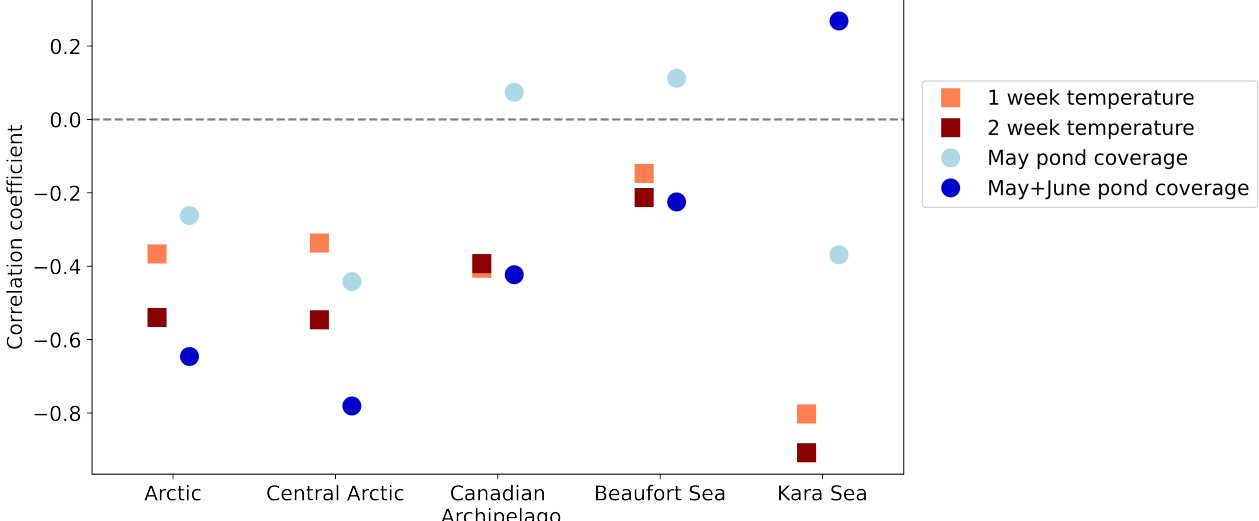

**Figure 12.** Correlation coefficients of September sea ice minimum extent with early summer melt pond coverage (circles) and with air temperatures just before the minimum (squares), shown for the entire Arctic and selected subregions. The correlation coefficients are calculated based on 7 years of data from 2017 to 2023. The sea ice minimum extent is averaged over the week before the observed minimum extent. The temperature is averaged over the same period of one week (light red squares) and two weeks (dark red squares). The melt pond covered area is accumulated for either May only (light blue circles) or May and June (dark blue circles).

predictor, but it can explain a part of the inter-annual variability of sea ice minimum extent and should therefore be taken into account when modeling this (Schröder et al., 2014; Liu et al., 2015). However, it is important to note that the major influence of melt ponds comes from the central Arctic region, while in the marginal seas the correlation is not decisive.

## 6 Conclusions


This study demonstrates the potential of the MPD2 melt pond fraction data product to analyze the seasonal evolution of melt ponds on local, regional, and pan-Arctic scales and investigate the fundamental interplay between melt pond fractions and environmental factors such as air temperatures and the sea ice surface topography. The latter is revealed by the subdivision of melt pond fraction observations based on the average distance between ridges of at least 20 cm in height from ICESat-2


data. The weekly inspection shows that higher melt pond fractions are observed first on level ice, which is likely because the melt water can easily spread over the sea ice surface and is not confined by ridges. Later, in July, the deformed sea ice with a high density of ridges tends to exhibit higher melt pond fractions, as the timing and extent of melt pond drainage gets more relevant. Melt ponds on the thinner, level ice often drain earlier, which reduces the melt pond fraction compared to the often thicker, deformed sea ice. Interestingly, this correlation could not be well reproduced by the analysis of melt pond fraction in


dependence on the sea ice type, i.e. multiyear and first-year ice, in 2023. Although ice type is a common approximation for



the degree of sea ice deformation. Here the melt pond fraction increases earlier on multiyear ice and also maximum melt pond fraction values for multiyear and first-year ice are similar. However, if integrated over the whole summer, melt pond fractions on first-year ice are still higher than for multiyear ice as can be expected from surface roughness.

The regional analysis by subdividing the Arctic into ten regions, i.e., the Central Arctic and the surrounding marginal seas, revealed strong regional differences in the seasonal evolution of melt ponds. The most striking influence can be attributed to the air temperatures. This is true for large-scale regional differences, as, for example, the Central Arctic, laying at higher latitudes, experiences the overall lowest melt pond fractions and temperatures. But also on smaller scales, an air temperature influence is found. This has been demonstrated by the comparison with in-situ observations for the MOSAiC expedition, where the MPD2 product can be used to observe short-term changes in melt pond fraction due to warm air intrusions. The highest melt pond fractions are consistently observed in the Canadian Arctic Archipelago, where the large areas of level landfast ice lead to an early, quick and widespread expansion of melt water on the sea ice surface. Other marginal Arctic seas often show higher seasonal variability in melt pond fractions. This high variability can be related either to higher sea ice drift activity, or to a strong reduction in total sea ice area over the summer. Also the impact of the presence of melt ponds in early summer on the sea ice minimum extent in September shows strong regional differences. Early melt pond formation influences the total summer sea ice energy balance but can either lead to sea ice break up or compact thin ice depending on other external forces. While the Central Arctic shows a strong anticorrelation between the melt pond coverage in May and June and the September sea ice minimum extent, this is reduced for the Canadian Arctic Archipelago and negligible for the Beaufort and Kara Seas. In the latter, in contrast, the sea ice extent is strongly influenced by the air temperatures immediately prior to the sea ice minimum.

Despite the overall high seasonal and inter-annual variability, a slightly increasing trend in the melt pond coverage has been detected within the analyzed time period 2017 to 2023. However, the time period is too short to draw further reaching conclusions from this.

The high temporal and spatial melt pond variability, found here for all studied cases, has to be taken into account when calculating surface energy balances for Arctic summer sea ice. The demonstrated regional dependencies of melt pond fraction on surface air temperature and surface topography can be useful for integrating melt ponds in climate models. However, for surface topography information about ridge distributions has to be available, which nowadays mostly is not the case for climate models.

*Data availability.* The Sentinel-3 satellite data used, is publicly available under https://ladsweb.modaps.eosdis.nasa.gov/archive/allData/450/ (last access August 14, 2023).

The aggregated MPD2 melt pond and open ocean fraction data with 12.5 km spatial resolution is available under https://data.seaice. uni-bremen.de/MeltPonds-Albedo/MPD2/.

ERA5 data are made available by the Copernicus Climate Change Service (C3S) at https://cds.climate.copernicus.eu/cdsapp#!/dataset/ reanalysis-era5-single-levels?tab=form.

The OSI-405-c sea ice drift data is available via https://thredds.met.no/thredds/catalog/osisaf/met.no/ice/drift_lr/merged/catalog.html.

The ASI sea ice concentration data is available via https://data.seaice.uni-bremen.de/amsr2/asi_daygrid_swath/n6250/.





The ICESat-2 drag coefficient data including the obstacle spacing as an approximation for the sea ice roughness is available on PANGAEA: https://doi.pangaea.de/10.1594/PANGAEA.959728.

    The in-situ transect melt pond fractions from the MOSAiC campaign have been provided personally by Melinda Webster.

*Author contributions.* HN has analyzed the data, and wrote the first draft of the paper. All authors developed the concept of the study, have actively contributed to discussions of the results and provided critical feedback on the paper.

*Competing interests.* The authors declare that they have no conflicts of interest.

*Acknowledgements.* We gratefully acknowledge the funding by the Deutsche Forschungsgemeinschaft (DFG, German Research Foundation) through the Transregional Collaborative Research Centre TRR-172 "ArctiC Amplification: Climate Relevant Atmospheric and SurfaCe Processes, and Feedback Mechanisms (AC)3" (grant 268020496) and the European Union's Horizon 2020 project CRiceS (grant 101003826). This research wouldn't have been possible without the availability of open data from satellites by space agencies ESA, NASA, and JAXA as
well as the in situ data from the MOSAiC expedition.



**Figure A1.** Seasonal evolution of melt pond fraction compared between the 7 years of Sentinel-3 data for the Chukchi Sea, the East Siberian Sea, the Laptev Sea, the Barents Sea, the East Greenland Sea, and the Baffin and Labrador Seas. The color indicates the daily averaged melt pond fraction per region following the color scale on the right. Gray areas represent data gaps. The vertical black lines mark the beginning and the end of the time where regionally averaged temperatures are above $0\,C°$. The light blue triangles indicate the initial melt pond formation defined by melt pond fractions passes the uncertainty of the MPD2 product of 8 %. The red circles highlight the time of the regional maximum melt pond fraction per year and the dark triangles the day when 75 % of the maximum value are exceeded.



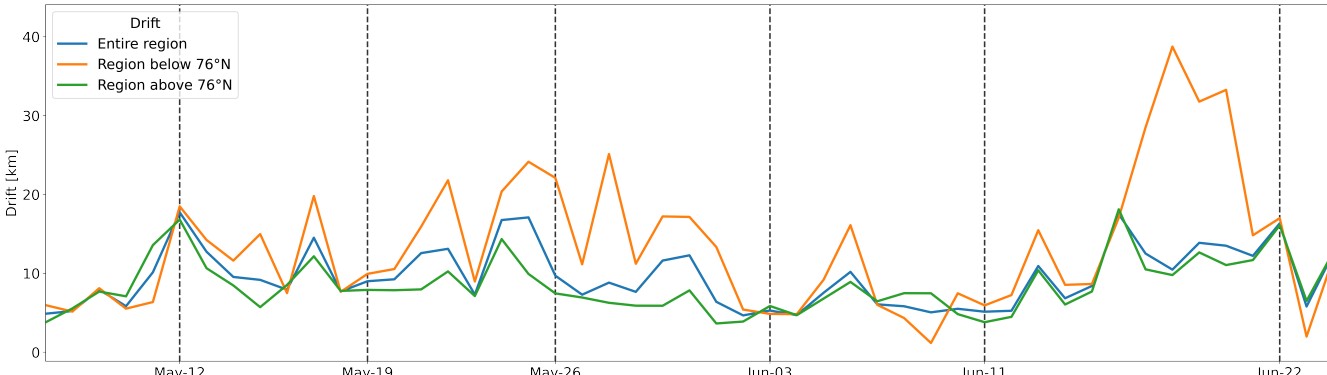

**Figure A2.** Sea ice drift from the OSI-SAF drift product averaged for the entire region of the Kara Sea (blue), the Kara Sea below 76 °N (orange), and the Kara Sea region above 76 °N (green). Shown are the data for the same period of year 2023 as presented in figure 9.



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
