# Peer review of "Regional and seasonal evolution of melt ponds on Arctic sea ice"

_EGUsphere, 2024_

## Author Response (AR1)

**Reviewer 1**

**We thank the reviewer for her/his insightful and helpful comment. We reply to them one by one below. The comments are in blue, our replies in black font.**

Review of "Regional and seasonal evolution of melt ponds on Arctic sea ice"

Tracking of melt pond coverage with satellite remote sensing remains the only viable method for full Arctic coverage across the melting seasons and through years. Given the importance of melt ponds to the surface energy balance of the Arctic Ocean, efforts to improve accuracy and coverage in said application are welcomed. Here, the authors present an effort to compile a 7-yr dataset of melt pond coverage from Sentinel-3 OLCI, representing an important step forward from prior studies involving MODIS as the EO instrument, as MODIS is now nearing the end of its life.

In the analysis, I particularly noted the results from surface roughness (ridge) impacts on melt pond fractions, which were interesting and novel. Overall, I feel that the paper carries sufficient weight to be considered for publication in TC. Some parts of the analysis could use improvement, with details listed in the comments below.

Thank you for this positive evaluation and your suggestions, which we comment on below.

**Major comments:**

Surface topography effects on MPF retrieval: Not only clouds but also surface topography features such as large and small-scale ridges and other ice deformities can cause shading of the surface at low Sun elevations. Have you considered if this can play any noticeable effect on the retrieval of MPF, which relies on the variability of the surface reflectance as the principal input? What are the satellite/illumination geometry limits you apply?

You are right also ridge shadows can cause uncertainty of the MPF retrieval. However, compared to cloud shadows the area affected by ridge shadows is small. The spectrum of shadows on snow/ice also is different from melt ponds. But indeed, on a satellite footprint scale of 1 km as used here, the spectral mixing of the shadows with other surfaces can cause an increase in MPF in the retrieval. We added a brief discussion about this uncertainty in the context of the cloud shodows. We also included some satellite geometry and background information in the beginning of Section 2.1.

Also, what is the general MPF retrieval uncertainty in MPD2, I assume you have an order of magnitude number available somewhere but this manuscript did not seem to

contain it? Considering the significance of the results w.r.t. observational uncertainties is of course always advisable in remote sensing studies, but perhaps that was treated in more detail by Niehaus et al. (2023)?

You are correct, the Niehaus et al. (2024) paper describes the retrieval including uncertainty in more detail. Here we focus on the geophysical interpretation of the results. Nevertheless, it is important to show the uncertainty here as well and we added a sentence at the beginning of Section 2 repeating the main results of the uncertainty estimates from Niehaus et al. (2024). Compared to higher resolution Sentinel-2 data we there estimate a RMSD of 7.8% and a bias of +1.6% for the MPF.

Temporal & spatial inhomogeneity filter: To be clear – all data over the 7-yr period is removed over the areas where the 0.25 threshold is crossed in the temporal side?

Yes.

The text is a bit ambiguous on this point.

You are right. It is not clearly enough mentioned that the full 7-year period 2017-2023 is used here. We added that in lines 116-121.

On the spatial filtering side, a similar question arose – you removed a full 7-day period if less than 20% of the region was covered by valid retrievals? Is the 20% condition thus different from the spatial homogeneity condition, where a threshold of 0.1 seems to apply (but was that value only for Central Arctic)? Please consider rewriting this section for clarity, I had quite some trouble following along.

Okay, good to know that the description of the spatial filtering is not clear enough. We reformulated the paragraph.
Yes, the 20% threshold is different from and applied in addition to the inhomogeneity measure. If the sea ice area observed in May in a particular region gets reduced to less than 20% at the end of the melt season, these days are removed from the time series. If we wouldn't do that the MPF of a very small area of remaining ice would strongly influence the calculated statistics (one could argue that even 20% already is a pretty low threshold).

Figure 7: Whereas Fig 6 has almost too much content to properly keep track of, this is a very good and informative figure, thanks for including it!

Thank you. We understand that Figure 6 contains a lot of information and also that a sideway figure is not ideal. We couldn't find a better way to present time series for all regions and still allow comparisons between the regions (which would be hard if we would split the figure).

Section 5 – the analysis is interesting and references to earlier such studies are appropriate. I was missing a more direct quantitative comparison to Schröder's and Liu's studies – could you not compute the correlation coefficients in a consistent manner with them to perform a S3-simulation and S3-MODIS intercomparison here?

We think that we did that. Schröder and Liu report a correlation of -0.8 for May and -0.7 to -0.8 for May+June, respectively, with sea ice minimum area for the whole Arctic. The left columns "Arctic" or "Central Arctic" in Fig. 12 provide our correlation values for these months. One has to keep in mind that the three studies cover different time periods (in addition to the different dataset/model).
We added a sentence in the text pointing out this difference but also explicitly mentioned the correlation coefficients of the three studies for easier comparison.

**Minor comments (line):**

195: "It would then fall into the third group that is otherwise contains" – I could not follow this sentence, please revise.

We did and specified for clarification that we talk about groups of Arctic regions.

218: "little deformed" -> meaning what?

We mean that the landfast sea ice usually is rather flat and shows not much ridges i.e. sea ice deformation.
We replaced the misleading "little" by "only slightly deformed"

379-380: Are there definite thresholds applied to classify large and small obstacle spacing in the text?

We added that information to the text (>1200m vs. <300m).

**Citation**: https://doi.org/10.5194/egusphere-2024-3127-RC1

**We thank the reviewer for her/his insightful and helpful comment. We reply to them one by one below. The comments are in blue, our replies in black font.**

**General Comments:**

This study uses the Melt Pond Detection 2 algorithm to examine the seasonal evolution and regional differences in Arctic melt pond fractions. It also examines the drivers of melt pond formation and evolution and the implications of melt ponds on the sea ice minimum. Overall, the manuscript is well-written, has a thorough methodology, and has sufficient results to support its conclusions. I particularly appreciate the comprehensive discussion of regional differences, which includes summaries that highlight the differences. The study effectively demonstrates the potential of the MPD2 product to analyze the seasonal evolution of melt ponds at various scales and their influence on sea ice variability and trends. As highlighted in the manuscript, analyzing changes in melt pond fraction during moist intrusion events is another valuable application of this product. Below, I provide several comments aimed at improving clarity and suggestions for enhancing the Figures.

Thank you very much for this positive evaluation and your suggestions.

**Specific Comments:**

*Figure 4:* Does a similar figure appear somewhere on NSIDC? If so, consider referencing that figure, moving Figure 4 to a supplement file, or adding the region outlines to Figure 2. The closest I could find is Figure 3 of "Special Report 25" at https://nsidc.org/data/nsidc-0780/versions/1, although it is less accessible and not as visually appealing.

Yes, as you say there is a map also at NSIDC but we think it is important to also show the regions within this article to have at at hand when talking about the regions a lot. We reduced the size a bit so that it doesn't take up as much space but would keep it in the article.

*Figures 5–7:*

- While it is clear that Figures 5–6 show the median melt pond fraction, it is unclear if Figure 7 shows the spatial mean or average melt pond fraction. "Average melt pond fractions" also appear in several places; does "average" consistently refer to the median, or is the spatial mean sometimes used? Clarification of this distinction would enhance reader understanding.

Thank you for the comment. We agree that the nomenclature should be unambiguous and we disentangled the terms "average", "mean" and "median" as they got mixed up a bit.

- Is the median used because it is more representative of the regional melt pond fraction than the mean given the skewed distributions and data filtering/masking?

Yes, exactly. The MPF distributions of a region can be very skewed and, in some cases, thus the mean can be misleading. The filtering/masking does not play a major role for that.

- Regional differences in melt pond fractions are somewhat challenging to discern in Figures 5–6. In contrast, regional differences are more apparent in Figure 7 and A1, and the summary in Table 1 is helpful. Have the authors considered plotting the 7-year average (mean or median) seasonal melt pond fraction? This could be added as an additional subplot in Figures 5–6 or an extra row in Figure 7. A 7-year average might further highlight regional differences.

Thank you for the suggestions. We think that is a good idea and would nicely summarize the regional differences. We added an extra row with the 7-year mean for each region in Figure 7 and A1.

*Line 274:* Is the uncertainty of 8% estimated using data over the entire Arctic, and is there evidence that certain regions experience substantially different retrieval uncertainties? It would also be helpful to cite Niehaus et al. (2024) here (or in the Figure 7 caption).

This uncertainty estimate indeed is from Niehaus et al. (2024) and we added that citation. The uncertainty is estimated based on higher resolution Sentinel-2 images from different regions but certainly not "the entire Arctic". We did not encounter different uncertainties for the regions as they are used here. But they can be higher in dynamic regions like the marginal ice zone. We also added a sentence about the uncertainty estimate at the beginning of Section2 where we introduce the data product.

*Surface energy budget:* Have the authors considered explicitly illustrating the impact of melt pond fraction on surface fluxes? While I appreciate the albedo comparisons in Section 4.1 and the correlations between accumulated melt pond-covered area and sea ice minimum extent, a visual comparison would be insightful. For instance, an additional row of subplots in Figure 9 showing albedo could provide a clearer link.

Albedo is also an output of the MPD2 retrieval. We added an extra row of albedo maps to Fig. 9 and a short paragraph discussing how both open water and melt ponds influence the albedo.

*Data availability:* Thank you for including links to all the datasets used in the study, particularly the gridded MPD2 melt pond fraction data. While the methodology is thorough, making some code publicly available would improve transparency and facilitate reproducibility.

Thank you for the suggestion and we agree that also open code sharing can help the scientific discussion. Nevertheless, the MPD2 method is based on a combination and heritage of C code and Python scripts from different authors and we do not feel that we can make that publicly available in a meaningful way. We will discuss if it makes sense to publish the code to reproduce the figures.

**Technical Corrections:**

*Line 56:* Change "temporal and and spatial" to "temporal and spatial."

We did.

***Line 190:*** **Change "the the Barents Sea" to "the Barents Sea."**

We did.

*Line 470:* Change "surface surface energy budget" to "surface energy budget."

We did.

**Citation**: https://doi.org/10.5194/egusphere-2024-3127-RC2